# Can Early Neuromuscular Rehabilitation Protocol Improve Disability after a Hemiparetic Stroke? A Pilot Study

**DOI:** 10.3390/brainsci12070816

**Published:** 2022-06-22

**Authors:** Mahdi Yazdani, Ahmad Chitsaz, Vahid Zolaktaf, Mohammad Saadatnia, Majid Ghasemi, Fatemeh Nazari, Abbas Chitsaz, Katsuhiko Suzuki, Hadi Nobari

**Affiliations:** 1Faculty of Sport Sciences, University of Isfahan, Isfahan 81746-7344, Iran; v.zolaktaf@spr.ui.ac.ir (V.Z.); abas.chitsaz@gmail.com (A.C.); 2Isfahan Neurosciences Research Centre, Alzahra Research Institute, Isfahan University of Medical Sciences, Isfahan 81839-83434, Iran; 3Isfahan Neurosciences Research Centre, Alzahra Research Institute, Department of Neurology, Isfahan University of Medical Sciences, Isfahan 81839-83434, Iran; chitsaz@med.mui.ac.ir (A.C.); saadatnia@med.mui.ac.ir (M.S.); ghasemimajid59@yahoo.com (M.G.); 4Isfahan Neurosciences Research Centre, Department of Adult Health Nursing, Faculty of Nursing and Midwifery, Isfahan University of Medical Sciences, Isfahan 81839-83434, Iran; nazari@nm.mui.ac.ir; 5Faculty of Sport Sciences, Waseda University, Tokorozawa 359-1192, Japan; 6Faculty of Sport Sciences, University of Extremadura, 10003 Caceres, Spain; 7Department of Motor Performance, Faculty of Physical Education and Mountain Sports, Transilvania University of Brașov, 500068 Brașov, Romania; 8Department of Exercise Physiology, Faculty of Educational Sciences and Psychology, University of Mohaghegh Ardabili, Ardabil 56199-11367, Iran

**Keywords:** acute stroke, early rehabilitation trial, exercise therapy, motor function

## Abstract

Background: The impairment of limb function and disability are among the most im portant consequences of stroke. To date, however, little research has been done on the early reha bilitation trial (ERT) after stroke in these patients. The purpose of this study was to evaluate the impact of ERT neuromuscular protocol on motor function soon after hemiparetic stroke. The sample included twelve hemiparetic patients (54.3 ± 15.4 years old) with ischemic stroke (n = 7 control, n = 5 intervention patients). ERTwas started as early as possible after stroke and included passive range of motion exercises, resistance training, assisted standing up, and active exercises of the healthy side of the body, in addition to encouraging voluntary contraction of affected limbs as much as possible. The rehabilitation was progressive and took 3 months, 6 days per week, 2–3 h per session. Fu gle-Meyer Assessment (FMA), Box and Blocks test (BBT) and Timed up and go (TUG) assessments were conducted. There was a significantly greater improvement in the intervention group com pared to control: FMA lower limbs (*p* = 0.001), total motor function (*p* = 0.002), but no significant difference in FMA upper limb between groups (*p* = 0.51). The analysis of data related to BBT showed no significant differences between the experimental and control groups (*p* = 0.3). However, TUG test showed significant differences between the experimental and control groups (*p* = 0.004). The most important finding of this study was to spend enough time in training sessions and provide adequate rest time for each person. Our results showed that ERT was associated with improved motor function but not with the upper limbs. This provides a basis for a definitive trial.

## 1. Introduction

Stroke is one of the most common causes of mortality and long-term disability in the world [1,2]. Fifteen million strokes occur worldwide every year, and about 90 million stroke patients currently live in different countries throughout the world, and this figure is anticipated to double by 2050 [3]. Disability and motor dysfunction are some of the main consequences of stroke that lead to 51% of the deaths occurring in the first 30 days following a stroke [4].

Countless rehabilitation techniques have been proposed for stroke-induced motor dysfunction, with one of the latest being an early rehabilitation trial (ERT) technique, which focuses on out-of-bed activities and has given researchers evidence to assume that it is associated with better health [5] and promise of walking [6] and reduced medical costs [7]. Additionally, patients have shown a greater desire for participation in early mobility programs than the standard post-stroke care protocols and research has been indicative of reduced mortality and fewer complications of immobility following these programs [8]. Nevertheless, there is little information about effective ERT and recovery strategies.

One way of increasing mobility immediately after a stroke is to use an exercise train ing or ERT protocol. This technique has attracted a lot of attention in the research commu nity because it has been shown to speed up recovery and eliminate certain stroke compli cations in animal models [9,10]. Exercise rehabilitation minimizes post-stroke motor and cognitive dysfunctions and thus appears to be a promising technique for stroke recovery [11]. The role of factors such as the time of beginning and duration of each session, and the activities and the intensity and type of protocols used for patients immediately after stroke remain to be further studied [12,13,14,15,16]. Most related studies on human models exam ining the period immediately after a stroke have been conducted either within or after the first two weeks of stroke [15,17] and no long-term interventions have been given to the samples tested. Furthermore, studies have frequently included a wide range of stroke se verities, from mild to severe, and no studies were found to perform a long-term interven tion on patients with moderate to severe complications immediately following a stroke. The present study was therefore conducted to investigate the effects of a three-month ex ercise rehabilitation program immediately after a stroke on patients’ performance.

## 2. Materials and Methods

### 2.1. Participants

The initial assessment of 700 stroke patients led to 75 eligible candidates, and 29 of them were then selected after further assessments and upon obtaining their consent. Av erage time from stroke to the start of the intervention and study, 44 h (control= 43.2 h, intervention= 44.6 h). The participants were then divided into an experimental and a control group using convenience sampling. The classification according to the specific physical condition of the patients was such that the researcher, after aimouncing his/her readiness to participate in the study, asked them the following question along with the explanations and placed them in the desired group. "In which group do you want to vol unteer for this research project? By the end of the study, one patient had died and four had withdrawn from the experimental group, and four had died and eight had withdrawn from the control group, meaning that the study was concluded with only 12 patients. The study’s flow is depicted in Figure 1. This study follows the statute of the CONSORT.

### 2.2. Sample Size

We used t-tests to evaluate the power and sample size for the design based on the statistical method examined with Power (1- err prob)= 0.80, based on past research on the influence of exercise on variables which revealed a large to very large effect size [18]. With a total of 12 subjects, there is an 83 percent (actual power) chance of properly reject ing the null hypothesis of no difference in variables. Also, for the second analysis of this study, we calculated sample size based on the t-tests - Correlation: Point biserial model; Power (1- err prob)= 0.95 and the same mentioned effect size. With a total of 8 subjects, there is an 83 percent actual power) chance of properly rejecting the null hypothesis of no difference in variables. G-Power software was used to conduct the statistical analysis (University of Dusseldorf, Dusseldorf, Germany).

### 2.3. Patient Demographics

Seven of the remaining patients were in the control group (two women and five men) and had a mean age of 47.4 ± 6.6 years, and five were in the experimental group (one woman and four men) and had a mean age of 64 ± 19.7 years. Of the 12 participating patients, nine were male (75%) and three were female (25%), and their mean National Institutes of Health Stroke Scale (NIHSS) score was 13.2 ± 3.0 and their mean MRS score was 4.58 ± 0.51 (Table 1).

All the participants completed written consent form before performing the tests and the stages of the research were approved by the ethics committee at the Isfahan University of Medical Sciences prior to commencement (NO: IR.MUI.REC.1394.2.084). The patients who met the inclusion criteria were assigned to either the training group or the control group based on their availability to participate in training.

### 2.4. The Experimental Approach to the Problem

The present quasi-experimental single-blind study was conducted in hospitals that provided specialized stroke services. The study subjects consisted of hospitalized stroke patients in Isfahan who were selected through convenience sampling. The study inclusion criteria were: (i) age over 18, a con-firmed first-time ischemic stroke; (ii) at least one-sided hemiplegia; (iii) alert and awake; (iv) a NIHSS score >8 (moderate to severe); (v) a modi fied ranking scale (MRS) score of 4–5; (vi) no other interfering diseases, (vii) a maximum of 72 h for the doctor to confirm the patient’s fit for inclusion in the study and personal and family consent for participation. The exclusion criteria were: confirmed transient is chemic attack (TIA), concomitant acute cardiovascular diseases, the condition exacerbat ing, relative recovery or abated symptoms after two weeks, the loss of consciousness and unwillingness to continue cooperation.

### 2.5. Neuromuscular Rehabilitation Protocol

The experimental group took part in ERT after entering the study (Table 2). The train ing sessions were held twice a day during the patient’s hospitalization period and were performed by a therapist; after discharge, the exercise protocol was performed three times per week by the therapist (e.g., the first two weeks every day by the therapist) and three additional times by the family/caregiver (previously, training was done as much as pos sible). The total time of each session was about 2 to 3 h, depending on the patient’s condition, we have 60 to 90 minutes of quality and useful and included exercises for both the healthy side and the affected side. The healthy side exercises (like resistance training) were performed with a greater intensity and power to boost strength and prevent muscu lar atrophy and bilateral transfer to the affected side (e.g., used hand held weights, Thera Band: 2 sets and 15–30 repetitions, RPE: 11–20). Passive exercises were performed to pre vent joint stiffness and reduce range of motion (ROM). He/she leaves the bed once he has attained the following abilities and with the assistance of others: standing test, step in place, squat, walking, and walking over obstacles. The patients were placed in a supine position with the top of the bed raised by 15–30 degrees. The coordination exercises were innovative and began from the very first days (with or without assistance) using finger control exercise, opening and closing the fingers, inward and outward rotation of the fin gers and passing a tennis ball (everything they can take and practice with, like a bottle of mineral water and so on) from one hand to the other to achieve coordination in line with the patient’s progress. The main points in the experimental group protocol were time of each exercise session and that maximum voluntary contractions were started in the ex tremities on both the healthy and affected sides as soon as the program began, and not using electrotherapy modalities. The patients in the experimental group were not allowed to take part in any physiotherapy programs during the intervention (three months) and their only therapy was the neuromuscular rehabilitation program. No specific interven tions were performed for the control group and they received only routine rehabilitation for stroke patients for three months.

### 2.6. Outcome Measures

At the end of the intervention and three months later, all the participants underwent reliability and validity test for motor function assessment, including the Box and Blocks Test (BBT) [19] of upper extremities, the Fugl-Meyer Assessment (FMA) [20] motor per formance (of upper & lower extremities, and the Time Up and Go (TUG) [21,22] of lower extremities. The results were assessed and compared between the two groups.

### 2.7. Statistical Analysis

The data collected was analyzed in SPSS-22 (IBM Co., NY, USA) at the significance level of *p* ≤ 0.05. The normal distribution of the data was assessed using the Kolmogorov Smirnov test, and the simple and accessible sampling were selected. The multivariate analysis of variance (MANOVA) was evaluated and age is applied as a covariate variable. Then, LSD post-hoc tests applied. Partial eta-square (qp2) was tested to find the magnitude of the differences (trivial effect <0.2, small effect 0.2, medium effect 0.5, large effect 0.8 and above). Independent t-test was also used to show the differences between the groups. G-Power software was used to conduct the statistical analysis for sample size (University of Dusseldorf, Dusseldorf, Germany).

## 3. Results

The results of the MANOVA analysis showed that there was a significant difference between participants’ performance based on group (F = 6.10, *p* = 0.026, 11p2 =0.80). The anal ysis of the data related to the FMA led to three groups of data: (A) The upper extremity; (B) the lower extremity; and (C) the total FMA score. Post hoc analysis showed, there were significant differences between dependent subjects in the variables of FMA lower extrem ity test (F = 26.7, *p* = 0.001, 11p2 = 0.75), total FMA score (F = 9.57, *p* = 0.013, 11p2 = 0.52) and TUG test (F = 18.9, *p* = 0.002, 11p2 = 0.68). In contrast, there were no significant differences between dependent subjects in the variables of FMA upper extremity test (F = 3.72, *p* = 0.09, l’}p2 = 0.29) and BBT test (F = 0.15, *p* = 0.71, l’}p2 = 0.02). The results of between groups are shown in Figure 2.

The analysis of the data related to the upper extremity in the FMA upper extremity test showed no significant differences between the experimental (35.6 ± 18) and control (13.57 ± 4) groups (*p* = 0.51), but the analysis of the data related to the FMA lower extremity test showed a significant difference between the experimental (29.4 ± 4.9) and control (15.8 ± 5.2) groups (*p* = 0.001), and the analysis of the data related to the total FMA score ulti mately showed significant differences between the experimental (65 ± 19.4) and control (29.4 ± 9.2) groups (*p* = 0.002).

The analysis of the data related to the BBT showed no significant differences between the experimental (9.6 ± 18.3) and control (0) groups (*p* = 0.3), but the TUG test showed significant differences between the experimental (23 ± 6.3) and control (51.1 ± 16) groups (*p* = 0.004).

The information on individual test evaluations before and after the neuromuscular rehabilitation protocol can be seen in Table 3.

In Figure 3 demonstrated the 95% confidence interval for the FMA upper extremity (ER: 28.2 to 43.0) vs. (SC: 11.7 to 15.4) and the FMA lower extremity (ER: 27.4 to 31.4) vs. (SC: 13.7 to 18.0).

## 4. Discussion

Very few studies have been conducted on the effectiveness of different programs on upper extremity recovery in the acute phase of stroke. The present findings on this subject concur with the results obtained in previous studies. Hubbard et al. demonstrate that early, more-intensive, upper-limb exercise improves motor functions [23]. Using modern robotic technologies, Stefan et al. demonstrated that early rehabilitation in the acute phase of stroke contributes to functional recovery in the upper extremity [24]. In an Effectiveness of Virtual Reality Exercises in Stroke Rehabilitation (EVREST) study, intensifying exercise whether through virtual reality or through simple games led to the early recovery of mo tor function in the upper extremity [25]. Although upper limb trend increased during this study. However, the present study shows that intense, early and long-term sports reha bilitation showed no significant differences between the experimental and control groups.

The studies conducted on lower extremity functioning suggest the positive effects of various early post-stroke mobility techniques on functional recovery. In two studies, pa tients with a mild stroke performed aerobic exercises immediately after their stroke and their walking speed, balance and endurance improved [25]. The results obtained in a pri mary outcome phase 3 a very early rehabilitation trial (AVERT) and an active mobility very early after stroke (AMOBES) study showed that early and intense activity has no significant effects on the patients’ mobility compared to routine care [17,26]. The present findings disagree with the results of most previous studies conducted in the acute phase of patients with moderate to severe stroke [13]. Since we showed that early, intense and long-term exercise rehabilitation has a very positive effect on functional abilities and walking, such that all the patients in the experimental group had the minimum ability to walk after three months, while only two of the patients in the control group gained this ability. Nonetheless, due to different gender in sample size, the results should be inter preted with care. Thus, the advantage of the ERT training program over previous pro grams, such as neural exercise training, determines the mechanisms of neural rehabilita tion. The intensity of rehabilitation appears to be effective in stimulating recovery through brain plasticity [17], how-ever most research has been done with patients in the subacute phase, from 2 weeks to 3 months. Few studies have been conducted on intensive basic training in humans, and conclusions from animal studies can be challenging because there are significant differences between animal models and individuals.

One last thing that needs to be carefully assessed is the timing, severity and type of the activities. The present study paid particular attention to the time of beginning the ac tivities, and given the sampling criteria and the results of previous studies, time was found to be closely associated with other factors, such as the patients’ state of conscious ness and comorbid diseases. When activities are started earlier, their effectiveness in terms of recovery and the reduction of complications is increased [27,28], and the present study supports the same results obtained in the ERT [29] regarding the start of activities, with the difference being that the present study also emphasizes the importance of other factors involved.

Severity, as discussed in previous studies, severity is a relative concept [17] that is defined and interpreted according to the patient’s conditions in each exercise session and during the course of treatment. Previous studies conducted on animal models showed that early and intense exercise worsens ischemia stroke and causes brain damage [30,31]. Recent studies, however, have demonstrated that early and intense exercise improves post-stroke functioning in rodents [32,33]. Human studies have found that intense activity does not positively affect the patients’ recovery more significantly than mild activity [28,32,33,34]; there have been even reports of intense and early activity exacerbating the complications and causing other disorders in patients with moderate to severe stroke [17,27]. Nonetheless, Imura et al. argued that early and intense rehabilitation significantly affects the patients’ functional recovery without increasing the complications of stroke [35]. Jennifer et al. reported that early and high-intensity stepping training (70–85% HRmax) improves walking and balanced outcomes in the stroke survivors [36]. Another study conducted on patients with mild stroke reported that early and intense aerobic ex ercise increases the walking speed and balance [14,25]. The present study sought to main tain a high intensity of activity in all the patients according to their readiness and partic ular conditions and used maximum voluntary contractions in both the healthy and the affected extremities to achieve this purpose, and gave alternate rest periods to the patients in order to prevent exhaustion, and eventually achieved favorable outcomes in the pa tients who completed the course of the treatment.

The type and amount of activity were one of the factors less emphasized in previous studies, because we found in only two trails that researchers used intensive aerobic exer cises regularly in 1- and 3-month intervention. In both studies, an improvement was re ported in the mild stroke patients’ walking speed, endurance, and balance [25]. In a study conducted by Di et al., which covered the first two weeks after stroke, a severe course of rehabilitation given two hours per day was reported not to affect the patients’ recovery much more than 45-minute rehabilitation [37]. A very early constraint-induced movement during stroke rehabilitation (VECTOR) study showed an intense course of rehabilitation given three hours per day to have negative effects on the patients’ recovery [38], and in the closest study to the present one that used 45-minute sessions of exercise in the first two weeks of stroke, no significant differences were reported in the outcomes compared to the results achieved with routine rehabilitation [17]. Secondary outcomes phase 3 ERT has shown that frequent short sessions out of bed are related to the best outcomes.

However, when too many long sessions are held, the chances of a desirable outcome are reduced and may even be harmful [39]. A meta-analysis in animals found that a shorter interval between stroke and exercise starts reduced infarct volume [40]. Neither of the studies had used this type of intervention (exercise rehabilitation plan) in a cross-sec tional form for several months; yet, all of the studies emphasized regular physical activity to prevent the complications of immobility [41,42]. The duration of the intervention and the regular exercise program offered to the patients are two advantages of this study (i.e., three 60 to 90 minutes vs. 2 to 3 h sessions per week for three months). The review by Schneider confirmed if we increased the time for each session to 100 minutes, the im proved outcome [43]. An observational study by James et al. designed to assess the time spent undertaking physical activity in physiotherapy sessions has demonstrated that pa tients spend 32 minutes of this time active [44]. In addition, the data from each exercise session showed that the differences resulting from the different types of rehabilitation given are noticeable even in the short-term, and the mobility test performed at the start and end of each exercise session confirms this point. Also, the actual power of this study for sample size can be another one of the strengths of the present study. In similar studies, articles have often been published before with less than 10 subjects [18,45,46].

In addition to factors including timing, severity, type and amount of exercise, other issues can also affect the results: (i) The consciousness and cooperation of the patients during exercise; (ii) The maximum use of voluntary contractions in both the healthy and the affected extremities over a specified period [47], and (iii) the patient’s medical history and pre-stroke lifestyle, as people with highly active lifestyles were more interested in taking part in early post-stroke mobility programs and had a faster process of recovery (and the factors of age and economic status were therefore not the only factors at play).

According to a study by Tanaka et al., Researchers state that physical activity and muscle contraction in the acute phase of stroke prevent muscle atrophy and decreased muscle mass. Due to the fact that muscle strength is related to muscle volume and mass, the lack of reduction in muscle mass in the process of early recovery of patients is directly related (standing and walking independently). Hence, they emphasized the importance of exercise programs in the acute phase [48]. Also, a study by Dromerick et al., states that early mobility in the form of motor exercises in the subacute group had the greatest effect on patients’ recovery compared to the control group [49]. However, in this study sampling of patients in the acute phase was considered ≤30 days after the stroke, while according to the latest classification of stroke phases, the acute phase is the first week after the stroke [50]. When most patients were discharged from the hospital, it could not be considered an acute phase. However, we started the study immediately after the first day of rehabilita tion.

Ultimately, according to the results obtained from the present research and previous studies, exercise and physical activities appear to prevent or postpone disability and lead to the patients’ recovery by way of two mechanisms: (i) cognitive mechanisms, including the many positive effects of physical activity on the spirit and mind and vitality and also the direct effects of exercise on physical fitness factors, releasing the mind of its treatment mode and inducing a sense of liveliness [3,51], and (ii) physiological mechanisms, including increased blood circulation, blood pressure control, increased oxygen tolerance, im proved neurogenesis and angiogenesis, mitochondrial viability, inhibition of the acute in flammatory response, neuronal apoptosis and protection of the blood-brain barrier [14,32,52].

The main study limitations, which significantly affected the cooperation of the pa tients, can be divided into five areas: First, the novelty of the rehabilitation method pro posed, which was very vague for the patients and their families and even the research team. Second, interventions are carried out as soon as possible after the stroke, and high intensity exercises are used. Third, the non-use of physiotherapy protocols in the course of the intervention in the experimental group and uncertainty about the treatment out comes. Fourth, the strict criteria for the inclusion of patients. Fifth, the lack of participation spirit in research project in the study community. Another limitation of the study is the significant age difference between the two groups which may include some personal, psy chological, and social confounding differences. Therefore, in future studies, we strongly urge researchers to consider this. However, this is one of the first studies in the Middle East that can provide useful information to target communities. Finally, we strongly rec ommend that future studies attempt a rehabilitation intervention with a larger number of participants.

## 5. Conclusions

The current study demonstrates that intensive, early, and long-term exercise rehabil itation improves total FMA score, FMA lower extremity and TUG tests. For this reason, it seems that based on this evidence, a comprehensive neuromuscular rehabilitation proto col such as ERT can be considered a modern technique for the rehabilitation of stroke patients. According to the results, the most important factor that was not considered in previous studies in the acute phase and the golden time of rehabilitation is the time of each session and the use of maximum voluntary contractions in the golden time after stroke. However, to confirm the findings, a larger trial should be performed on a wider range of patients. The researchers hope that in the near future, these results will help to develop a modern and safe technique to help stroke patients achieve their independence in the shortest possible time.

## Figures and Tables

**Figure 1 brainsci-12-00816-f001:**
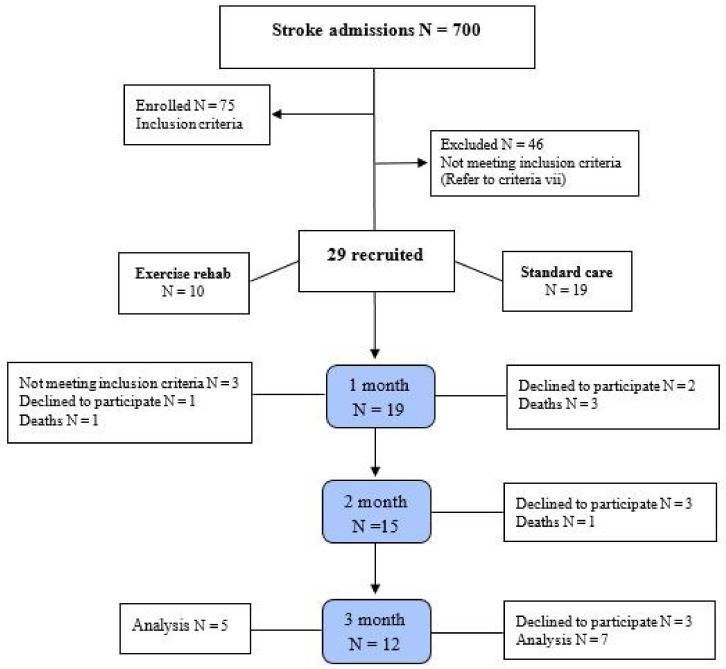
CONSORT flow chart of participants for recruitment, application, follow-up and analysis.

**Figure 2 brainsci-12-00816-f002:**
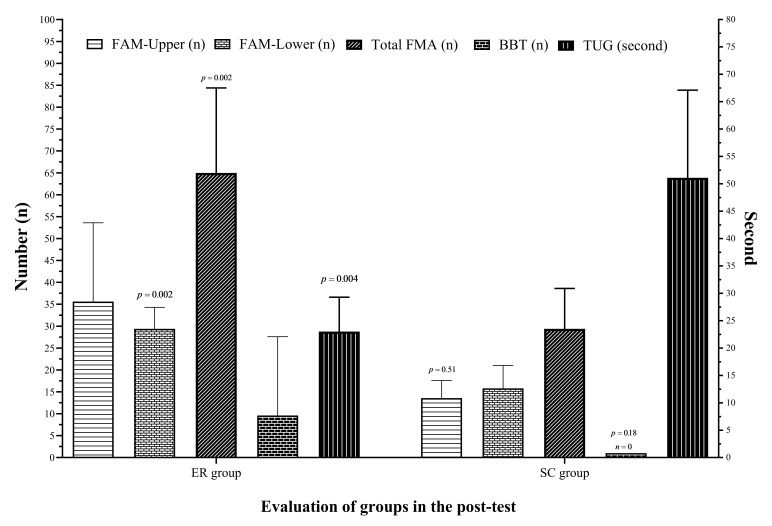
Comparison of groups in assessments. ER = Exercise rehabilitation, SC = Standard care, FMA = Fugle-Meyer Assessment, BBT = Box and Blocks Test, TUG = Time Up and Go.

**Figure 3 brainsci-12-00816-f003:**
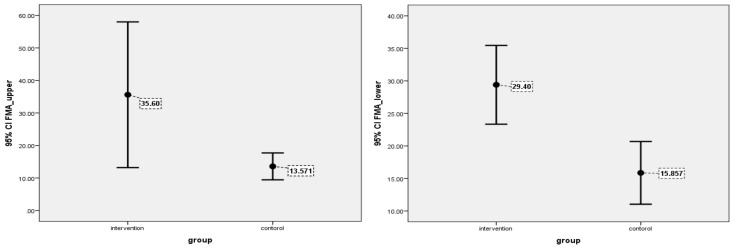
Error bars plots showing result of Fugl-Meyer Upper & Lower function between two groups.

**Table 1 brainsci-12-00816-t001:** Patient demographics.

Role	Group	Sex	ge	Lesion Location	NIHSS	MRS	LS	TTS
1	ER	Female	36	Frontoparietal + basal ganglia	12	4	left	72
2	ER	Male	81	Pons + medulla	10	4	Right	48
3	ER	Male	83	basal ganglia	9	4	Right	24
4	ER	Male	67	Frontoparietal + basal ganglia	16	5	left	48
5	ER	Male	53	Basal ganglia + temporal cortex	11	5	left	24
6	SC	Male	42	Tempofrontal+ basal ganglia	16	5	left	24
7	SC	Male	53	Basal ganglia	13	4	Right	48
8	SC	Female	51	Frontoparietal + basal ganglia	13	5	left	72
9	SC	Female	40	Frontopariental + basal ganglia	17	5	Right	24
10	SC	Male	57	Basal ganglia	17	5	left	72
11	SC	Male	48	basal ganglia	16	5	left	24
12	SC	Male	41	Basal ganglia	9	4	left	48

ER = Exercise rehabilitation, SC = Standard care, NIHSS = National institutes of health stroke scale, MRS = Modified ranking scale, LS = Lesion side, TTS = Time to start.

**Table 2 brainsci-12-00816-t002:** Neuromuscular rehabilitation protocol.

Mode of Exercise	Major Goals	Intensity/Frequency/Duration
Flexibility Stretching (ROM upper & lower limb)	• Increase ROM of involved extremities • Prevent contractures	• 6 d/wk • ROM stretch for 10 minutes
Strength • Circuit training • Weight machines • Free weights • TheraBand • Isometric exercise Aerobic exercise (15–30 min)	Increase independence in ADLs	• 2 sets of 15–30 repetitions of 8–14 exercises • RPE: 11–20 (6–20 scale) involving the major muscle groups • heart rate: under maximum, 220-age *•* 6 d/wk
Neuromuscular Coordination and balance activities • Use everything they can take and practice with	• Improve level of safety during ADLs	6 d/wk

Type of training: Fle & ext elbow (WWR), Fle & ext shoulder (WWR), Horizontal Fle & ext shoulder (WWR), Abd & add shoulder (WWR), Fle & Ext leg (WWR), Abd & add leg (WWR), Bridging, knee Fle (prone (Ext, (WWR), and plantar & dorsiflexion (WWR). After getting at least abilities and with the support he leaves the bed: Standing test, step in place, squat, walking, walking over obstacles. The given program emphasized voluntary contractions and work on the acute phase of stroke and was modified according to each patient’s personal conditions and abilities. WWR: with and without resistance, RPE: rating of perceived exertion, ROM: range of motion, and ADL: activity of daily living.

**Table 3 brainsci-12-00816-t003:** Results of effectiveness measures for each participant.

Role	Groups	FAM Upper (n)		FAM Lower (n)		TUG (s)		BBT (n)	
		Pre	Post	Pre	Post	Pre	Post	Pre	Post
1	ER	3	18	5	34	0	14.35	0	0
2	ER	4	31	6	26	0	29.57	0	0
3	ER	29	64	5	30	0	25.76	0	42
4	ER	0	24	4	23	0	26.77	0	0
5	ER	2	41	10	34	0	18.61	0	6
6	SC	0	8	2	10	0	60 *	0	0
7	SC	2	14	4	15	0	60 *	0	0
8	SC	2	11	2	16	0	60 *	0	0
9	SC	2	16	3	18	0	37.46	0	0
10	SC	0	12	2	9	0	60 *	0	0
11	SC	0	12	0	19	0	60 *	0	0
12	SC	0	22	0	24	0	20.24	0	0

ER = Exercise rehabilitation, SC = Standard care, FMA = Fugl-Meyer Assessment, BBT = Box and Blocks Test, TUG = Time Up and Go, 60 * = Each patient who was unable to walk received a score of 60.#.

## Data Availability

The data used to support the findings of current study are available from the corresponding author upon request.

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
