# Peer review of "Can Early Neuromuscular Rehabilitation Protocol Improve Disability after a Hemiparetic Stroke? A Pilot Study"

_brainsci, 2022, doi:10.3390/brainsci12070816_

Round 1

Reviewer 1 Report

The manuscript is well drafted and relevant. 

Author Response

The manuscript is well drafted and relevant. 

Dear reviewer, we thank you for your positive comment.

Reviewer 2 Report

Dear Editor,

The manuscript entitled: “Can Early Neuromuscular Rehabilitation Protocol Improve Disability After a Hemiparetic Stroke? A Pilot Study” provided to assess impact of early rehabilitation trial neuromuscular protocol on motor function following hemiparetic stroke. There are comments which needs to be noted.

  1. The grouping method are not explained. In line 103,” … were assigned to either the training group or the control group based on their availability to participate in training”, does it mean participants of each group was selected arbitrarily without considering significant age difference (17 years) between two groups which brings up age related personal, psychologic, and social confounding interventions?
  2. In spite of non-significant difference in upper extremities in FMA and BBT, how authors claim that “rehabilitation has positive effects on upper extremity recovery” (lines 197 to 199)?
  3. In line 206, reference # 17 discussed improvement in patients’ mobility and reference # 26 only studied upper-limb. How authors refereeing these studies for claiming lack of recovery in mobility and lower-limb?

Based on authors claim on reference # 17, Neurological Exercise Training (NExT) Program inefficiency on mobility recovery, thus it’s needed to discuss ERT exercise program advantage on prior programs such as NExT and the possible determining neuro-rehabilitative mechanisms.

  1. The explanation in Line 208 to 212 is not convincing. because of age-related variation between groups, it is required to show results of effectiveness measures for each participant.
  2. Manuscript has grammar errors and needs language improvement.
  3. Line 99: Table 2 or 1?
  4. Line 121: Table 2 or 1?
  5. Numbering repeat: 2.4.
  6. Figure 2 legends missed.

Author Response

Dear Editor and Reviewers,

Thank you very much for your kind and valuable comments. All changes in the manuscript

were highlighted in yellow.

Kind regards

The authors

Reviewer 1

The manuscript is well drafted and relevant. 

Dear reviewer, we thank you for your positive comment.

Reviewer 2

Dear Editor,

The manuscript entitled: “Can Early Neuromuscular Rehabilitation Protocol Improve Disability After a Hemiparetic Stroke? A Pilot Study” provided to assess impact of early rehabilitation trial neuromuscular protocol on motor function following hemiparetic stroke. There are comments which needs to be noted.

Dear reviewer, we thank you for your valuable comments and all your comments were considered word by word in the article and all of them are answered below.

  1. The grouping method are not explained. In line 103,” … were assigned to either the training group or the control group based on their availability to participate in training”, does it mean participants of each group was selected arbitrarily without considering significant age difference (17 years) between two groups which brings up age related personal, psychologic, and social confounding interventions?

Authors Answer: Dear reviewer, thank you so much for your attention. We have added the following information in the manuscript. However, we do agree with your point about the age related personal, psychologic, and social confounding interventions, for that reason we also added this point as a limitation of study and suggestion for readers and researchers to consider for future study. We hope you have a thorough understanding of the difficulties of gathering information and the interest of participants in such studies. An important challenge in the more than two years that we have had in gathering information on this project, and most importantly, this is one of the first studies in the Middle East that is unique in its kind.

Line 83-87: The participants were then divided into an experimental and a control group using convenience sampling. The classification according to the specific physical condition of the patients was such that the researcher, after announcing his/her readiness to participate in the study, asked them the following question along with the explanations and placed them in the desired group. "In which group do you want to volunteer for this research project?"

Line 140-146: The study subjects consisted of hospitalized stroke patients in Isfahan who were selected through convenience sampling. The study inclusion criteria were: (i) age over 18, a confirmed first-time ischemic stroke; (ii) at least one-sided hemiplegia; (iii) alert and awake; (iv) a NIHSS score >8 (moderate to severe); (v) a modified ranking scale (MRS) score of 4-5; (vi) no other interfering diseases, (vii) a maximum of 72 hours for the doctor to confirm the patient’s fit for inclusion in the study and personal and family consent for participation.

Line 332-336: Another limitation of the study is the significant age difference between two groups which may include some personal, psychologic, and social confounding differences. Therefore, in future studies, we strongly urge researchers to consider this. However, this is one of the first studies in the Middle East that can provide useful information to target communities.

  1. In spite of non-significant difference in upper extremities in FMA and BBT, how authors claim that “rehabilitation has positive effects on upper extremity recovery” (lines 197 to 199)?

Authors Answer: Dear reviewer, thank you for your careful reminder. We mentioned the result of the correlation between the two in this section, which was not correct in writing. We corrected it here and also corrected the report in the results section and the correlation presentation section.

  1. In line 206, reference # 17 discussed improvement in patients’ mobility and reference # 26 only studied upper-limb. How authors refereeing these studies for claiming lack of recovery in mobility and lower-limb?

Authors Answer: Dear reviewer, the reference 17 is related to the AMOBES article and a software error has been made. Now it is correct. You can see reference in below. Regarding 26 reference, you are right, we have omitted.

Yelnik AP, Quintaine V, Andriantsifanetra C, Wannepain M, Reiner P, Marnef H, Evrard M, Meseguer E, Devailly JP, Lozano M, Lamy C, Colle F, Vicaut E; AMOBES Group. AMOBES (Active Mobility Very Early After Stroke): A Randomized Controlled Trial. Stroke. 2017 Feb;48(2):400-405. doi: 10.1161/STROKEAHA.116.014803. Epub 2016 Dec 22. PMID: 28008092.

Based on authors claim on reference # 17, Neurological Exercise Training (NExT) Program inefficiency on mobility recovery, thus it’s needed to discuss ERT exercise program advantage on prior programs such as NExT and the possible determining neuro-rehabilitative mechanisms.

Authors Answer: Dear reviewer, we have added the relevant information in two different sections of the discussion in lines 252 to 258 as well as lines 224 to 334.

  1. The explanation in Line 208 to 212 is not convincing. because of age-related variation between groups, it is required to show results of effectiveness measures for each participant.

Authors Answer: Dear reviewer, we added the new table based on individual information.

  1. Manuscript has grammar errors and needs language improvement.

Authors Answer: thank you and the article checked by native speaker person.

  1. Line 99: Table 2 or 1?

Authors Answer: done.

  1. Line 121: Table 2 or 1?

Authors Answer: done.

  1. Numbering repeat: 2.4.

Authors Answer: corrected.

  1. Figure 2 legends missed.

Authors Answer: added.

Reviewer 3 Report

The manuscript entitled “Can Early Neuromuscular Rehabilitation Protocol Improve Disability After a Hemiparetic Stroke? A Pilot Study” is aimed to assess the impact of ERT neuromuscular protocol on motor function post hemiparetic stroke.

The authors have given data to prove effectiveness of their exercise-based rehab protocol post-stroke and explained them but still there are a few more issues which needs to be addressed before publication.

Major issues

  • Showing the results in table 3 in graphical format will be more appropriate.
  • Results part explaining the Figure 2 (last paragraph) needs to be rewritten as in present form it does not sufficiently explain.
  • Figure title missing for figure 2
  • Figure legends/description needed for both the figures.
  • Conclusion needs to be rewritten. Needs to explain the major findings very briefly before suggesting future ideas.
  • Language/sentence formation needs to be checked thoroughly. Some sentences like line 262-263 are not understandable.

Minor issues

  • Only department and institute names must be given in the author designation. Avoid job designation.
  • “Protocol determining” not a suitable keyword. Need better keywords.
  • Few grammar and spelling mistakes in the text (For eg. Line 82 “..atual..”)
  • Some sentences like line 88-90 should be shifted from “Materials and Methods” to Discussion section.
  • Punctuation errors in few places pertaining to the brackets closing.

Author Response

Dear Editor and Reviewers,

Thank you very much for your kind and valuable comments. All changes in the manuscript

were highlighted in yellow.

Kind regards

The authors

Reviewer 1

The manuscript is well drafted and relevant. 

Dear reviewer, we thank you for your positive comment.

Reviewer 2

The manuscript entitled “Can Early Neuromuscular Rehabilitation Protocol Improve Disability After a Hemiparetic Stroke? A Pilot Study” is aimed to assess the impact of ERT neuromuscular protocol on motor function post hemiparetic stroke.

The authors have given data to prove effectiveness of their exercise-based rehab protocol post-stroke and explained them but still there are a few more issues which needs to be addressed before publication.

Dear reviewer, we thank you for your valuable comments and all your comments were considered word by word in the article and all of them are answered below.

Major issues

  • Showing the results in table 3 in graphical format will be more appropriate.

Authors Answer: Dear reviewer, we have changed and added new figure based on your suggestion. Thank you.

  • Results part explaining the Figure 2 (last paragraph) needs to be rewritten as in present form it does not sufficiently explain.

Authors Answer: Dear reviewer, we have rewritten that paragraph.

  • Figure title missing for figure 2

Authors Answer: Added.

  • Figure legends/description needed for both the figures.

Authors Answer: Done.

  • Conclusion needs to be rewritten. Needs to explain the major findings very briefly before suggesting future ideas.

Authors Answer: We added main result of study and also we have rewritten conclusion.

  • Language/sentence formation needs to be checked thoroughly. Some sentences like line 262-263 are not

Authors Answer: We revised and also, the article checked by native speaker person.

Minor issues

  • Only department and institute names must be given in the author designation. Avoid job designation.

Authors Answer: Corrected.

  • “Protocol determining” not a suitable keyword. Need better keywords.

Authors Answer: thank you so much, we changed.

  • Few grammar and spelling mistakes in the text (For eg. Line 82 “..atual..”)

atual power = actual power

Authors Answer: Revised, thank you.

  • Some sentences like line 88-90 should be shifted from “Materials and Methods” to Discussion section.

Authors Answer: We do totally agree with your suggestion, we moved that sentences to suitable paragraph of the discussion.

  • Punctuation errors in few places pertaining to the brackets closing.

Authors Answer: We modified the text carefully. A person native speakers also examined it.

Round 2

Reviewer 2 Report

Grouping strategy based on volunteers will, could bring psychosomatic factors to intervene results of exercise-based programs. Hence, it is recommended to design future studies in blind way.